# Research of Water Absorption and Release Mechanism of Superabsorbent Polymer in Cement Paste

**DOI:** 10.3390/polym15143062

**Published:** 2023-07-16

**Authors:** Xiao Qin, Yongkang Lin, Jie Mao, Xiaolong Sun, Zhengzhuan Xie, Qingjian Huang

**Affiliations:** 1School of Transportation and Civil Engineering and Architecture, Foshan University, Foshan 528225, China; 2112161019@stu.fosu.edu.cn (Y.L.); 2112211006@stu.fosu.edu.cn (Q.H.); 2Guangdong GuanYue Highway and Bridge Co., Ltd., Guangzhou 511450, China; maojacker@126.com; 3School of Civil and Transportation Engineering, Guangdong University of Technology, Guangzhou 510006, China; xls1998@gdut.edu.cn; 4Guangxi Key Lab of Road Structure and Materials, Nanning 530007, China; xzz249817896@163.com

**Keywords:** superabsorbent polymer, internal cured cement paste, dynamic swelling behavior, solution properties, capillary negative pressures, water release performance

## Abstract

The water absorption and release properties of superabsorbent polymers’ (SAP) internal curing agent are affected by many factors, such as solution properties, the ambient temperature and humidity and the particle size of SAP, which determine the curing effect and the durability of cement concrete structures directly. In this paper, the variation rule of the water absorbing capacity of SAP in simulated cement paste under different solutions and environmental conditions was studied. Based on microscopic image technology, the dynamic swelling behavior of the SAP particles was explored. The water release performance of SAP in cement paste was analyzed by both the tracer method and the negative pressure method. The results show that the water absorption of SAP in cement paste varied from 27 to 33 times. The ionic valence had a significant effect on the water absorption capacity of SAP, which suggests that the larger the ionic radius, the lower the absorption of SAP. The higher the temperature of the solution, the greater the water absorption rate of SAP. While the SAP particle size was less than 40–80 mesh, a slight ‘agglomeration effect’ was prone to occur, but the absorption state of SAP was more stable. Based on the swelling kinetic equation of SAP and the time-dependent swelling morphology of SAP in cement paste, a swelling kinetic model was established. The water release performance of SAP was less affected by the capillary negative pressures, and it would not release the water prematurely during the plastic stage, which was conducive to the continuous internal curing process of hardened paste in the later stage.

## 1. Introduction

The phenomenon of microcrack initiation and insufficient hydration are significantly generated in the cement concrete structure of highway engineering that is exposed to the atmosphere and are most likely caused by the frequent evaporation of water and water consumption during the hydration process, thus increasing the risk of deterioration of the concrete structure during the service period. Thus, reasonable curing measures should be taken to inhibit the generation of humidity shrinkage cracks effectively [1]. At present, the material composition design of cement concrete tends to reach higher performance than before, which makes it difficult for curing water to infiltrate the material adequately due to its respectively dense structure. This will cause the frequent initiation of humidity shrinkage cracks, which may increase the risk of deterioration during the structure service period.

Cement concrete internal curing technology is one of the most promising technologies of shrinkage and crack resistance that can inhibit the humidity shrinkage cracks by releasing water via the internal curing agent that is incorporated into the cement paste [2]. Nowadays, superabsorbent polymers (SAP) are one of the best internal curing agents. The pre-absorbent SAP particles can release water in a timely fashion, while the humidity of the cement paste decreases and the ion concentration of the pore solution increases, which can effectively reduce the shrinkage strain, promote the hydration of cementitious materials and improve the mechanical properties and durability of cement concrete [3,4].

In effect, the development of a curing effect, microstructure properties and the macro-performance of the internal curing cement concrete mainly depends on the water absorption and release performance of the SAP. However, the water absorption of SAP always changes dynamically with the ion concentration and pH of the cement paste in the early stage of mixing [5]. The calculation error of internal curing water was often caused by the inaccuracy of the SAP water absorption test in previous studies. Too much or too little absorption and the absorbed rate will adversely affect the construction workability, mechanical properties and durability of cement concrete. Therefore, in order to control the water absorption and water release performance of SAP precisely and gain the most benefit from the internal curing effect of SAP in cement concrete, it is necessary to research the ‘water absorption-release’ mechanism of SAP in cement paste.

Nevertheless, most of the existing studies focus on the improvement of the macro-properties of cement concrete by SAP, including the mechanical property, shrinkage resistance, crack resistance and durability. Sun et al. [6] found that SAP can reduce the compressive strength of concrete under water-curing conditions, while SAP can improve the later compressive strength of concrete under air-curing conditions. Liu [7] and Kalinowski [8] further analyzed the effect of SAP on the compressive strength of concrete. They believe that in the early stage, the addition of SAP would form large pores and reduce the compressive strength of concrete. With the increase of age, due to the hydration products formed by the internal curing water provided by SAP to compensate for the pores, the microstructure near the SAP was densified and the mechanical properties were compensated.

In the study of water absorption and the release performance of SAP, Schröfl [9] and Kang [10] studied the water absorption kinetics of SAP and proposed that SAP absorption in a solution is mainly controlled by osmotic pressure, which could be regarded as a diffusion process and can be expressed by Fick’s second law. Bi [11] studied the short-term SAP absorption characteristics in cement paste and clarified the SAP absorption characteristics with the change of the dynamic chemical environment of water paste. However, the studies above only focused on the self-absorption of SAP. In addition, Yang [12] and Lee [13] found that the SAP absorption capacity was affected by the total ion concentration and concentration of [Ca^2+^] in the process of cement hydration. Girum [14] and Yun [15] proposed that the water absorption-release capacity of different SAP in cement paste depends on its crosslinking density and the density of anionic functional groups. Tan et al. [16] explored the SAP swelling characteristics in different solutions.

Li et al. [17] studied the water absorption and release behavior of SAP by using the squeezing pore solution and autogenous shrinkage test. Zhang et al. [18] have shown that the SAP water absorbed rate in a cement centrifugate increased rapidly, then gradually decreased and finally stabilized. Snoeck et al. [19] used nuclear magnetic resonance (NMR) to study the water release effect of SAP cementitious materials during hydration, and proposed water release kinetics that effectively alleviate autogenous shrinkage. Mehdi et al. [20] observed the water migration of cement-based materials during hydration by neutron radiography.

In conclusion, there is a certain amount of research on the water absorption-release performance of SAP in pure water and cement paste, but it mainly focuses on the simple study of water absorption performance. The existing research results have not yet involved the systematic study of the water absorption-release characteristics of SAP in fresh cement slurry, and related dynamic absorption-release mechanism research has not yet been carried out. Therefore, there is a lack of more scientific and reasonable methods to determine the amount of internal curing water, which causes the construction workability, mechanical properties and durability of the designed internal curing concrete to be unstable.

In this paper, the influence of environmental factors, such as the water–binder ratio, fly ash content, ionic valence, temperature and SAP particle size on the absorption performance of SAP was studied. Based on the swelling kinetics equation and image analysis technology, the absorption kinetics of SAP in cement paste were further studied. Moreover, the influence mechanism of temperature, humidity and capillary pressure on the water release behavior of SAP in cement paste was explored. Finally, the water absorption-release mechanism of SAP in fresh cement paste was proposed.

## 2. Materials and Experimental Methods

### 2.1. Materials and Mix Proportion

SAP consisting of irregular powdered sodium polyacrylate by aqueous solution polymerization was used as an internal curing agent and the molecular formula was (C_3_H_3_NaO_2_)_n_. In the synthesis, deionized water was used as a solvent; acrylic acid was used as a monomer; ammonium persulfate was used as an initiator; N, N-methylenebisacrylamide was used as a crosslinking agent; sodium hydroxide and sodium chloride were used as neutralization agents; the monomer concentration was 50%; and the neutralization degree was 70%. All chemical reagents from Shanghai Sinopharm Chemical Reagent Co., Ltd. in China. Acrylic acid was neutralized to sodium acrylate firstly, and then sodium acrylate was polymerized to sodium polyacrylate. The state and morphology of the SAPs are presented in Figure 1 and Figure 2, respectively. SAP was used with dry particle sizes of 20–40 mesh (380–830 μm, SAP-20), 40–80 mesh (180–380 μm, SAP-40) and 100–120 mesh (120–150 μm, SAP-100).

In this study, ordinary Portland cement (PO.42.5) produced by Guangdong Yingde Conch Cement Co., Ltd. was used with density of 3.15 g/cm^3^. The chemical composition of the cement is shown in Table 1. The fly ash was grade I produced by Guangdong Zhongye fly ash Co., Ltd. in China, and its activity index and loss on ignition were 75% and 4.95%, respectively. Polycarboxylate superplasticizer (JB-ZSC) produced by Guangdong Strong Official Building Materials Science and Technology Co., Ltd. in China with a water reduction rate of 26% and a gas content of 3.1% was employed.

The coarse and fine aggregates were eliminated to configure the cement paste closer to the internal ion concentration of the concrete by a large number of optimization experiments in the early stage of research. The detailed ratio can be seen in Table 2.

### 2.2. Simulated Cement Paste Solution Configuration

Due to the thick consistency of cement paste and the large number of particles, this study was based on the ion concentration measured in real cement paste and a high-precision configuration of various cement paste simulation solutions to research the absorption of SAP. The concentration of [Na^+^], [K^+^], [Ca^2+^], [OH^−^] and [SO_4_^2−^] and the pH of fresh cement paste after 30 min were measured by a PXS-215A digital ion concentration meter from Shanghai Yueci Electronic Technology Co., Ltd. in China. The solution temperature was 25°C ± 2°C, and a two-point positioning method was used. The test results are shown in Table 3. The simulated solution was prepared according to the measured ion concentration of the fresh cement paste. The chemical reagents used included CaSO_4_, Na_2_SO_4_, K_2_SO_4_, NaOH and KOH. The solvent was deionized water. The mix proportions of the simulated cement paste solution used in this study are listed in Table 4. In Table 3 through 6, 0.37–10% of 0.37 represents the water–binder ratio, 10% represents the percentage of fly ash in the cementitious material, and other values are the same as above.

### 2.3. SAP Water Absorption Test

In the simulated solution of cement paste, the teabag method was chosen for the water absorption test of SAP. The testing process is as follows: first, the teabag was soaked in the solution, and the mass (*m_t_*) was weighed after the water was added. Subsequently, the dry SAP with a mass of *m*_s_ was placed in another teabag, immersed in the solution until the swelling equilibrium was reached, and the teabag was taken out to discharge the water freely. After the water stopped dripping, the total mass m*_a_* was weighed, and the water absorption of SAP(*Q*) could be calculated by Equation (1).
(1)$$Q=\frac{m_{a}-m_{t}-m_{s}}{m_{s}}\times 100\%$$

### 2.4. Metallographic Microscope Test

In order to explore the swelling kinetics of SAP in the internal environment of cement concrete, a Phenix XZJ-L2030 optical microscope from Jiangxi Phenix Optical Technology Co., Ltd., in China was used to examine the SAP volume in the cement paste at different time points, and the Fick characteristic index was calculated to determine the dynamic swelling behavior of the SAP in the cement paste.

### 2.5. SAP Water Release Distance Tracer Test

According to the results of the water absorption test, the SAP gel absorbing the same ratio of red ink was placed in white cement paste. The *W/B* of the white paste was 0.31 and the fly ash content was 15%. After the cement paste was solidified, a microscopic observation instrument was used to observe the diffusion range of the internal curing water in the SAP to the cement paste at different ages (i.e., water release distance), as see in Figure 3. The greater the water release distance, the better the effect in the SAP.

### 2.6. SAP Water Release Test in Capillary Pressure Environment

The pore pressure of cement paste before the final setting (10 h) was continuously tested by a capillary hole pressure test device, and the results are shown in Figure 4. It can be seen that the pore pressure of the cement paste varied from 0–74.56 kPa in 0–10 h. Based on the measured data, the pore pressure at 10 kPa (initial setting), 25 kPa (6 h), 55 kPa (8 h) and 75 kPa (final setting) was simulated by vacuum equipment, and the water release rate of the SAP gel under different negative pressures of the capillary pores was studied.

## 3. Results and Discussions

### 3.1. SAP Absorption Responsiveness Analysis

#### 3.1.1. Influence of Solution Type

The test results of SAP absorption and the absorbed rate in different simulated cement paste solutions are shown in Figure 5. It could be seen that the water absorption of SAP in different simulated solutions was 27–33 times, and the absorption in the *W/B* = 0.37 simulated solution was slightly higher than that in the *W/B* = 0.31. Additionally, the SAP in different simulated solutions reached the overall absorption equilibrium state at about 30 min, but slight water release occurred after 30 min and reached equilibrium again at 60 min. Because the SAP initial absorbed rate is faster, the internal network structure expansion has acceleration, that is, the internal network fails to retract in time while it reaches the swelling equilibrium. While the actual absorbed rate decreased significantly, the internal network will have a ‘network closing effect’.

It is clear in Figure 5c that the absorption of the SAP in the *W/B* = 0.31 simulated solution increased with the decrease of the ion concentration during the initial stage of absorption, that was, 0.31–25% > 0.31–20% > 0.31–15% > 0.31–10%; however, the absorption showed an opposite variation rule during the slight water release stage. Nevertheless, the variation of SAP in the *W/B* = 0.37 simulated solution is not as obvious as the former, as can be seen in Figure 5a. Therefore, it is considered that SAP has better water retention in a lower water–binder ratio and a high ion concentration solution, which is conducive to the gradual release of SAP internal curing water for the internal curing effect.

It is worth noting that, except for the absorbed rate at 1 min, the absorbed rate of the SAP in the simulated solution of *W/B* = 0.37 was essentially the same (see Figure 5d). However, for the simulated solution with *W/B* = 0.31, the difference of absorbed rate in the first 10 min of SAP was large, and the variation rule was complex. The maximum absorbed rate was about 77.36% higher than the minimum absorbed rate (see Figure 5c). This phenomenon can be explained as follows: the multivalent ions [Ca^2+^], [OH^−^], [SO_4_^2−^] in the solution and the -COOH and [Na^+^] in the SAP gel produce the ‘common-ion effect‘. In the solution containing a variety of complex ions, the higher the ion concentration, the greater the ionization balance of the original electrolyte moves in the original direction, which lowers the binding probability of each ion to the -COOH and [Na^+^] in the SAP gel; therefore, the lower the ionization degree of the original electrolyte, and the more unstable the solution.

#### 3.1.2. Influence of Ionic Valence

The influence of the ionic valence on SAP absorption and the absorbed rate are presented in Figure 6. With the increase of the ionic valence of the solution, the SAP absorption was smaller, that is, NaOH > CaSO_4_ > 0.31–15%. The absorption of SAP in CaSO_4_ and 0.31–15% was almost equal at 1 min, but the two rapidly separated at 5 min. This is probably due to the short reaction time of the interaction between the SAP and various ions in complex solutions.

As seen in Figure 6b, the higher the ion valence of the solution, the lower the SAP absorbed rate. At the beginning of water absorption, there was a large difference in the absorbed rate among the three mixtures, but it gradually approached the same rate by 5 min. In addition, the SAP reached the water absorption equilibrium at 30 min in CaSO_4_ and 0.31–15%, where it was still active in the absorption state at 50 min in NaOH, and the absorbed rate showed an upward trend. These phenomena could be interpreted as follows: while the ion valence in the solution is higher, the water absorption and absorbed rate of SAP are lower; thus, the absorption equilibrium time of SAP is shorter, and the absorption state is more stable.

The reason for this is that the hydration between cations and water in the solution decreases with the increase of the ionic radius. The ionic radius of [Na^+^] is greater than that of [Ca^2+^]; therefore, the hydration of [Na^+^] is greater than that of [Ca^2+^], and the shielding effect of water on [Na^+^] is greater than that of [Ca^2+^], resulting in the osmotic pressure between SAP and NaOH being greater than that of CaSO_4_. Another reason is that [Ca^2+^] is easy to combine with the hydrophilic group -COO- of SAP to form a crosslinking point, which increases the crosslinking density and reduces the absorption. As a result, the absorption of SAP in NaOH was higher than that in CaSO_4_. The 0.31–15% contained a large number of multivalent anions and cations, the ‘common-ion effect‘ caused the osmotic pressure on the inner and outer sides of SAP to be lower, the degree of network structure contraction increased, and the absorption rate dropped sharply.

#### 3.1.3. Influence of pH

Previous experiments found that the pH of cement paste fluctuated between 12.30–12.55 before mixing and final setting, with an average of pH of 12.43. On this basis, pH = 12, pH = 12.43 and pH = 13 were chosen to study the absorption performance of SAP. The pH of the solution was adjusted by adding deionized water and NaOH. The deionized water was added to decrease the pH of the cement paste; NaOH could be added to increase the pH of cement paste. The test results are shown in Figure 7.

From Figure 7, it can be seen that the higher the solution pH, the smaller the SAP absorption and absorbed rate. In the cement paste with pH = 12.43 and pH = 13, the SAP reached the absorption equilibrium at 30 min, but it reached the equilibrium state at 40 min in the cement paste with pH = 12. Secondly, the equilibrium absorption of SAP in the cement pastes with pH = 12, pH = 12.43 and pH = 13 were 39.133 times, 33.399 times and 11.116 times higher, respectively. It could be seen that, while the pH of the cement paste was in the range of 12 to 12.43, it had little influence on the absorption of SAP, whereas with pH = 13, the absorption of the SAP was greatly reduced.

In general, the carboxylic acids will be converted into carboxylates while the SAP is in a moderately alkaline solution, and mutual repulsive forces will be generated between the carboxylates, which leads to expansion of the SAP and increased volume, thereby increasing the water absorption (see Figure 8). On the contrary, the absorption of SAP in cement paste did not satisfy this rule with the variation of pH. These phenomena could be mainly explained by three aspects: firstly, the ion effect occurs between various cations in cement paste, and the expansion effect of SAP is continuously reduced. Secondly, the concentration of [Na^+^] in the solution increases with the increase of pH, resulting in a decrease in the osmotic pressure of the SAP internal network, which has a negative effect on the absorption. Thirdly, the [Ca^2+^] and [Mg ^2+^] in the solution will combine with -COOH to form a hard complex, which increases the crosslink density degree of the SAP internal network, so that the network shrinkage is enhanced, and the absorption is reduced.

#### 3.1.4. Influence of Temperature

The construction temperature of cement concrete pavement engineering is quite different during different periods. Various factors, such as the ambient temperature and water temperature during construction, will have a certain impact on the water absorption performance of SAP. Furthermore, the cementitious material will produce a certain degree of hydration heat during the hydration process, which will also affect the absorption temperature and performance of the SAP in the cement paste. As a consequence, four temperatures of 10 °C (winter), 25 °C (spring and autumn), 35 °C (summer) and 60 °C (the hydration heat of pavement concrete often reaches 60 °C) were selected to study the water absorption of SAP. The test results are depicted in Figure 9.

It can be concluded from Figure 9 that SAP generally followed the law that the higher the temperature of a solution, the greater the absorption and absorbed rate, and the shorter the swelling equilibrium time of SAP gel. The reason is as follows: while the temperature is high, the motion unit of the SAP polymer chain is activated, which increases the motion energy of the SAP polymer chain. Until the energy is increased enough to overcome the energy barrier required by the motion unit to move in a certain way, the motion unit will be activated and begin to move thermally in a certain way, including the whole chain motion, chain segment motion, chain joint motion and other types, thus increasing the diffusion coefficient of the SAP gel network and accelerating the water absorption. Then, the volume expansion of SAP is large when the temperature is high, and there is significant free space between the molecules, which is a sufficient condition for the motion of various motion units. After the free space reaches the size necessary for the motion of a certain motion unit, the motion unit can move freely and quickly. In summary, SAP swelling is the diffusion process of its gel network in a solvent, and the relaxation and stretching process of SAP polymer chains are jointly promoted by the activation and volume expansion of polymer chain motion units. This is reflected by the fact that the higher the temperature of a solution, the faster the motion of the SAP polymer chain and the faster the network expansion, leading to higher absorption and absorbed rates.

As shown in Figure 9a,c, the equilibrium absorption of the SAP in 0.31–15% was 1.55%, 8.77%, 0.58% and 7.59% lower than that in 0.37–15%, respectively. These phenomena indicate that the motion unit of the SAP polymer chain is in an inert state, even if the ion concentration of 0.31–15% is higher than 0.37–15% at a low temperature (10 °C), but the difference between the two is not large due to the limited stretching ability of the SAP gel network. With the increase of temperature (25 °C), the motion energy of SAP increased slightly, and the stretching ability increased, thereby widening the absorption of SAP in two different concentration solutions. However, while the temperature of the solution increased to 35 °C, the thermal motion state of the SAP polymer chain reached the threshold, and the SAP gel network diffused rapidly during this acceleration period. The influence of temperature on the absorption is slightly greater than the ion concentration of the solution; therefore, the absorption of the SAP in the two concentration solutions is similar. The thermal motion state of the SAP polymer chain long exceeded the threshold value while the solution temperature was 60 °C. The influence of the solution ion concentration on its absorption is greater than the temperature of the solution, and the difference of the SAP absorption in the two solutions increased again.

On this basis, according to the Karadag superabsorbent polymer absorption equation [21], the measured absorbing data of SAP were fitted, and its expression is shown in Equation (2).
(2)$$\frac{dQ}{dt}=K_{Q}{\left(Q_{eq}-Q\right)}^{2}$$
where *Q* is absorption; *t* is the absorbed time; *K_Q_
*is the absorbed rate constant; and *Q_eq_* is the equilibrium absorption measured by the test.

From Equation (2), the integral Equation (3) can be derived:(3)$$\frac{t}{Q}=A+Bt$$
where *A* is the reciprocal of initial absorbed rate, *A* = 1/(*K_Q_Q_eq_*^2^); and *B* is the reciprocal of theoretical equilibrium absorption, *B* = 1/*Q_t_,_eq_*.

In order to verify the accuracy of the test results, the experimental data were fitted to obtain the absorption kinetic parameters, as seen in Table 5. The theoretical equilibrium absorption obtained by fitting was almost consistent with the experimental results, and the error did not exceed 5.5%. On the whole, the water absorption of SAP increases with the increasing temperature, which indicates that the fitting result is accurate, and the absorbed rate constant can reflect the average absorbed rate of SAP in the absorption process as a whole. Moreover, the absorbed rate constant did not completely follow the rule that the time in which the absorbed rate decreases to reach the swelling equilibrium becomes shorter with the increase of temperature, which is consistent with the variation of the absorbed rate curve (see Figure 9b,d). This conclusion shows that the temperature has a complex influence on the water absorption of SAP, and the increase of temperature can promote the absorption of SAP to a certain extent.

#### 3.1.5. Influence of SAP Particle Size

The variation rules of SAP absorption with different particle sizes are exhibited in Figure 10. As is described in Figure 10, during the initial stage of absorption, the absorption of SAP increased with the decrease of the particle size, that is, SAP-100 > SAP-40 > SAP-20. However, the absorption was opposite of that during the initial stage of absorption at 5 min. This is because the smaller the particle size of the SAP in the same mass, the greater the number and the larger the contact area with the solution, the shorter the distance of the solution from the SAP surface to the inside, and the shorter the time to reach the swelling equilibrium. In addition, it is common to observe a slight ‘agglomeration effect’ when the SAP particle size is less than 40–80 mesh and the surface has been swelled with water; however, the inner portion is still flocculent, which leads to the smaller a particle size of the SAP and a reduced equilibrium absorption.

It was also observed that the absorption of the SAP in 0.37–15% was higher than that in 0.31–15%, and the equilibrium absorption of SAP-20 in 0.37–15% and 0.31–15% was 75.82%, but in SAP-40, it was 95.67% and in SAP-100, it was 87.95%. This phenomenon shows that the smaller the particle size of the SAP, the smaller the influence of the solution concentration on water absorption, and the more stable the water absorption state, thus ensuring a step-by-step release of the curing water for the long-term internal curing effect. Therefore, a small particle size of the SAP is more suitable for internal curing of cement concrete.

### 3.2. Dynamic Swelling Behavior of SAP Based on Metallographic Microscopic Image Analysis

The volume of the SAP gel was determined by a grid method; it was assumed that the SAP gel particles are spherical particles, and the radius was calculated by the number of grids. According to the volume of the SAP gel in cement paste at different times, the swelling kinetics of the SAP were expressed by Equation (4) [22]:(4)$$\frac{Q_{t}}{Q_{eq}}=K_{s}t^{n}$$
where *Q_t_* is the absorption at *t* time; *Q_eq_* is the equilibrium absorption; *K_s_
*is the characteristic constant of the SAP gel; and *n* is the Fick characteristic constant, which is used to describe the swelling mechanism of SAP; it can reflect the relationship between the solvent diffusion rate and the polymer chain relaxation rate.

For the convenience of calculation, let *F* = *K_s_t^n^*; a linear equation is obtained by utilizing logarithms on both sides of the equation, as in Equation (5):(5)$$\ln F=n\ln t+\ln K_{s}$$

In accordance with Equation (5), the volume variation and water absorption could be calculated, and the swelling ln*F-*ln*t* curve could be obtained by the swelling process of SAP within 1–30 min in the image, as shown in Figure 11.

As can be observed from the figure, the correlation between the data points before 5 min and the fitting curve was low, but the correlation between the data points and the fitting curve was high after 5 min, and the swelling process of the SAP in cement paste was generally linear. The reason for this observation is that the first 5 min of SAP is the rapid period, the internal polymer chain movement is fast. The network expansion is rapid and the internal concentration difference changes greatly; therefore, the SAP presents an unstable swelling state, causing a low correlation between the data points before 5 min and the fitting curve.

The diffusion of solvents in SAP gels is mainly divided into three types [23]: (1) Fick diffusion, which is when the diffusion of the solvent satisfies the Fick diffusion law, and its Fick characteristic constant *n* ≤ 0.5; (2) relaxation equilibrium diffusion, which is a polymer chain relaxation control process, *n* ≥ 1.0; and (3) non-Fick diffusion, in which the solvent diffusion rate is comparable to the polymer chain relaxation rate, *n* = 0.5–1.0. This demonstrates that the Fick characteristic constants of SAP with different particle sizes in different cement pastes were less than 0.5. This explains that the water absorption behavior of the SAP in cement paste meets the Fick diffusion law, which mainly depends on the hydrophilicity of the gel, and the cement paste can freely diffuse into the gel.

Furthermore, the gel characteristic constants of the SAP in 0.37–15% were less than 0.31–15%, and the Fick characteristic constant was greater than 0.31–15%. In other words, the greater the ion concentration of a solution, the larger the SAP gel characteristic constant and the smaller the Fick characteristic constant, which could be due to the fact that the greater the ion concentration of a solution, the smaller the absorption of SAP, thus the shorter the free diffusion path of the cement paste into SAP and the shorter the diffusion time.

According to the calculation of Equations (4) and (5), the swelling kinetic model of SAP in cement paste is shown in Table 6.

### 3.3. Water Release Behavior of SAP in Cement Paste

#### 3.3.1. Influence of Temperature and Relative Humidity

The SAP gel will release water in the course of the changing temperatures and humidity of cement concrete. According to the construction temperature and relative humidity of cement concrete pavement at different times, 10 °C (winter), 25 °C (spring and autumn), 35 °C (summer) and 60 °C (hot) were used as the representative temperatures, and 0% RH (dry), 75% RH (conventional) and 90% RH (high humidity) were used as the representative humidity levels when the SAP water release capacity was tested. The water release trajectory of SAP at 25 °C for 3 days and 7 days is depicted in Figure 12. It can be seen that the SAP gel volume shrinks due to the water release. Figure 13 illustrates the test results of the SAP water release distance at different temperatures and relative humidity.

From Figure 13a, it can be seen that the water release distance of SAP increased with the increase of temperature within 7 days because the higher the temperature, the faster the water inside the cement paste evaporates, the greater the demand for water, and the more water is released by the SAP. In addition, the water release rate of SAP reached the fastest rate in the first 3 days, and then decreased significantly after 3 days. The main reason for the laws above may be the humidity shrinkage of cement paste, which mainly occurs in the first 3 days. Within this timeframe, the hydration reaction of the gel material is fast, and the water demand rate is large. The SAP has an obvious effect on the internal curing of cement paste. After 3 days, the hydration reaction rate of the cementitious material decreased to a stable stage, and the internal humidity of the cement paste reached equilibrium. The water demand rate also decreased; therefore, the water release rate of SAP slowed down significantly.

Figure 13b reveals that the water release distance of SAP decreased with the increase of relative humidity from days 1 to 7. The water release distance of 50%, 75% and 90% relative humidity at day 3 was 86.08%, 87.32% and 34.62% of that at day 7, respectively. Clearly, except for the high humidity environment, with a relative humidity of 90%, the SAP in the other two relative humidity environments had obvious internal curing in the first 3 days. It is indicated that SAP can supplement the water of cement paste in a timely manner, increase the internal relative humidity and reduce the early microfracture caused by humidity shrinkage when the relative humidity is low.

#### 3.3.2. SAP Water Release Performance of Cement-Based Materials under Capillary Negative Pressure

The water release rate of SAP in different simulated capillary negative pressures is presented in Figure 14. 

In Figure 14, under the capillary negative pressures from the initial setting stage to the final setting stage of cement-based materials, the water release rate of the SAP appeared to continuously increase from 2.164% to 14.115%. It could be concluded that the capillary negative pressure is not the main driving force of SAP water release. The water release performance of SAP is less affected by the capillary negative pressures, and it has a stable water retention capacity, which can avoid the phenomenon of excessive water release during the plastic stage of cement paste.

There is essentially no internal stress during the plastic stage of the cement paste, and its structural stability cannot be maintained. Therefore, the water released by the SAP can play an effective internal curing effect after the setting and hardening of the cement paste. Moreover, the cement paste has a higher relative humidity during the early stage and the relative humidity during the plastic stage is as high as 99%. If excessive water is released by the SAP during the plastic stage, it will affect the internal curing of the later hardening stage, resulting in insufficient shrinkage resistance. Consequently, during the practical application of the project, the water release of the SAP process was controlled to maintain a step-by-step method to achieve continuous and stable internal curing.

## 4. Conclusions

The influence of different solutions and environmental conditions on the water absorbing capacity of SAP and the dynamic swelling behavior of SAP were investigated in this paper. The water release behavior of SAP in cement paste was also researched to help us understand the enhancement mechanism of internal curing with SAP. The main conclusions are outlined below:The water absorption of SAP in different simulated solutions was 27–33 times, which demonstrates a stable absorption performance. In addition, the water retention performance of SAP was excellent in a high ion concentration solution.The influence of ion valence on the water absorbing capacity of SAP was mainly due to the different ion radii in the solution, which could suggest that the larger the ion radius, the smaller the osmotic pressure of the SAP internal network and the lower the water absorption of SAP.The absorption of smaller particle size SAP was related to the obstruction of the expansion effect in a strong alkaline solution. Moreover, the increase of temperature could accelerate the water absorption of SAP in the medium solution effectively.While the particle size of SAP was less than 40–80 mesh, a slight ‘agglomeration effect’ was prone to occur. The surface-absorbed water has been swelled, but the inner portion is still flocculent.The water absorption process of SAP mainly depends on the hydrophilicity action of the SAP gel, which conforms to Fick’s second diffusion law.The water release performance of SAP was mainly affected by the humidity difference and ion concentration difference, but was less affected by the capillary negative pressures.

## Figures and Tables

**Figure 1 polymers-15-03062-f001:**
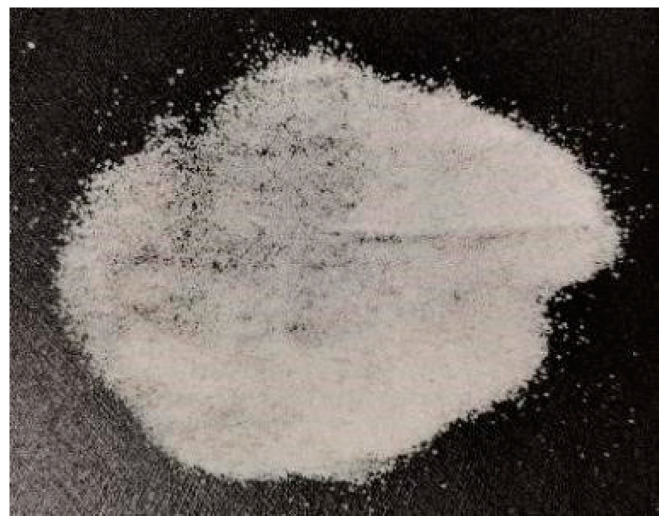
SAP in dry state.

**Figure 2 polymers-15-03062-f002:**
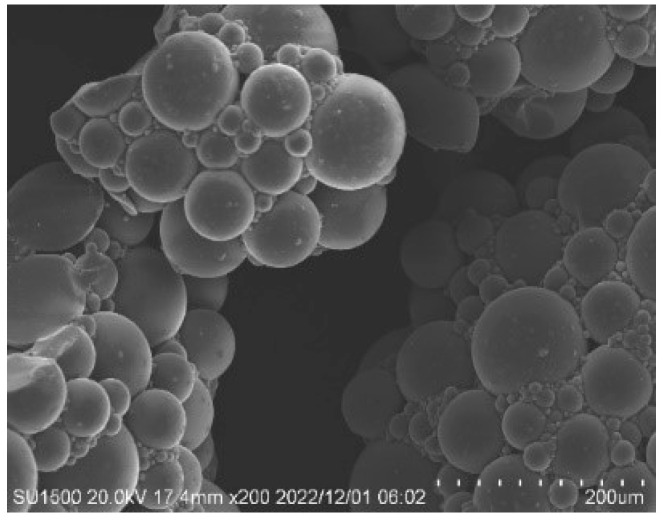
Microscopic morphology of SAP.

**Figure 3 polymers-15-03062-f003:**
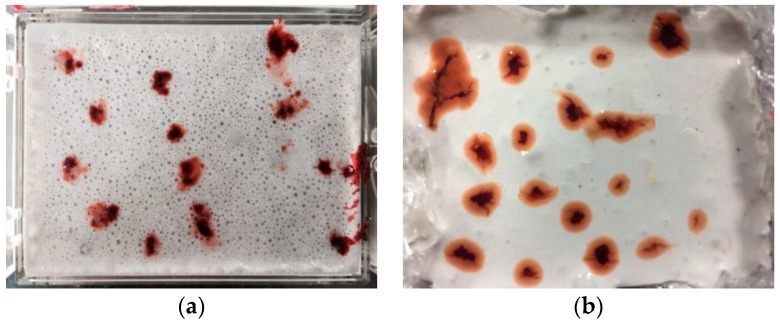
SAP water release distance tracer test: (**a**) initial state of SAP internal curing cement paste; (**b**) hardening state of SAP internal curing cement paste.

**Figure 4 polymers-15-03062-f004:**
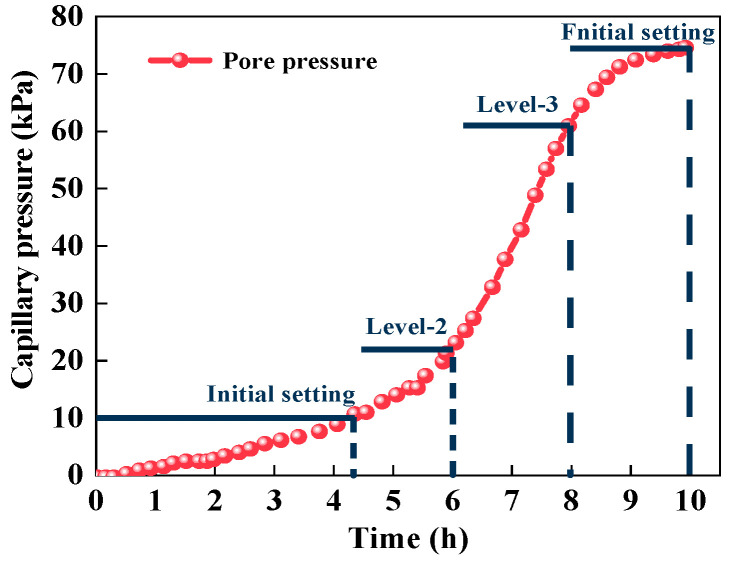
Pore pressure of cement paste.

**Figure 5 polymers-15-03062-f005:**
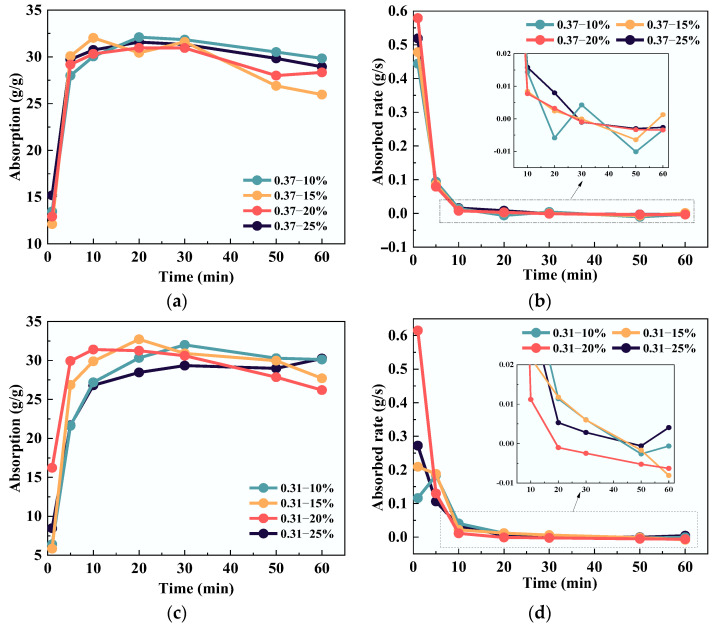
Influence of solution type: (**a**,**b**) *W*/*B* = 0.37; (**c**,**d**) *W*/*B* = 0.31.

**Figure 6 polymers-15-03062-f006:**
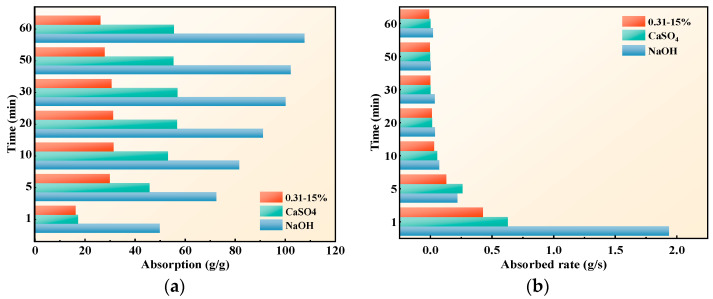
Influence of ionic valence: (**a**) absorption; (**b**) absorbed rate.

**Figure 7 polymers-15-03062-f007:**
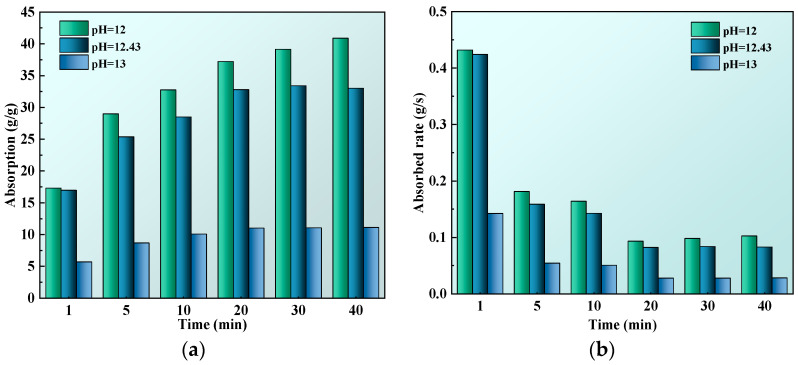
Influence of pH: (**a**) absorption; (**b**) absorbed rate.

**Figure 8 polymers-15-03062-f008:**
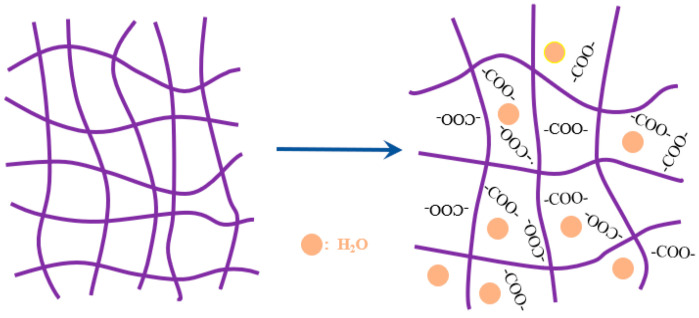
The expansion diagram of SAP in moderately alkaline solution.

**Figure 9 polymers-15-03062-f009:**
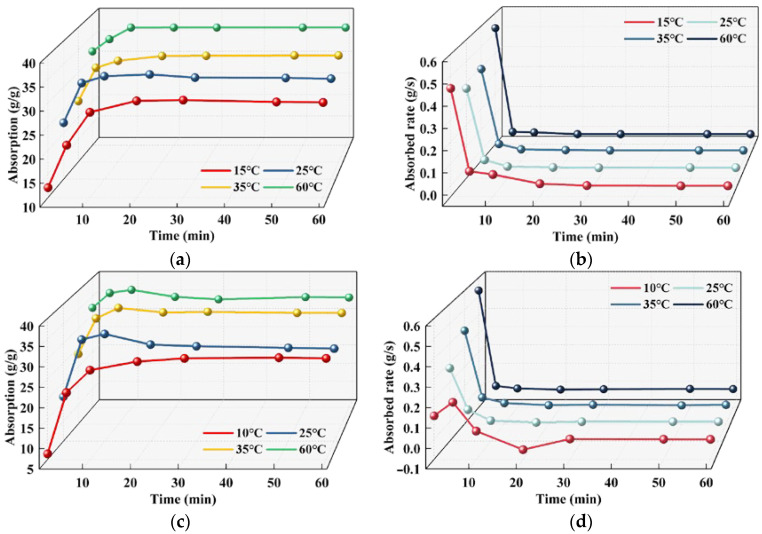
The influence of temperature on the water absorbing capacity of SAP: (**a**) absorption of 0.37–15%; (**b**) absorbed rate of 0.37–15%; (**c**) absorption of 0.31–15%; (**d**) absorbed rate of 0.31–15%.

**Figure 10 polymers-15-03062-f010:**
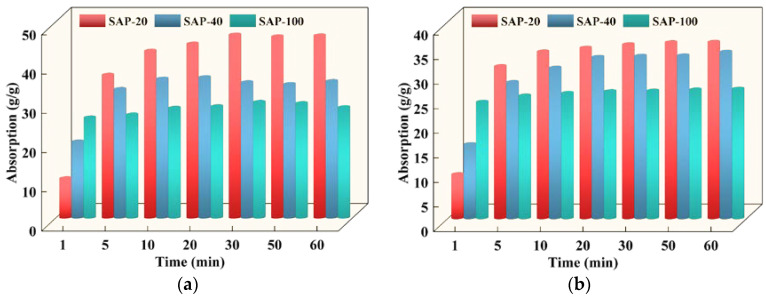
The absorption of different SAP particle sizes in different solutions: (**a**) 0.37–15%; (**b**) 0.31–15%.

**Figure 11 polymers-15-03062-f011:**
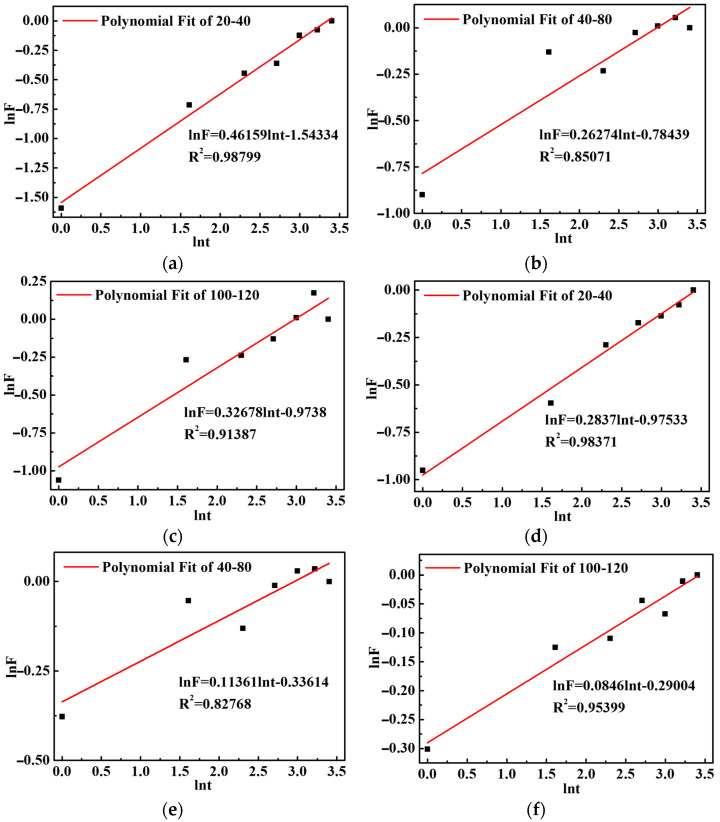
The swelling curves of SAP with different particle sizes in cement paste: (**a**–**c**) SAP in 0.37–15%; (**d**–**f**) SAP in 0.31–15%.

**Figure 12 polymers-15-03062-f012:**
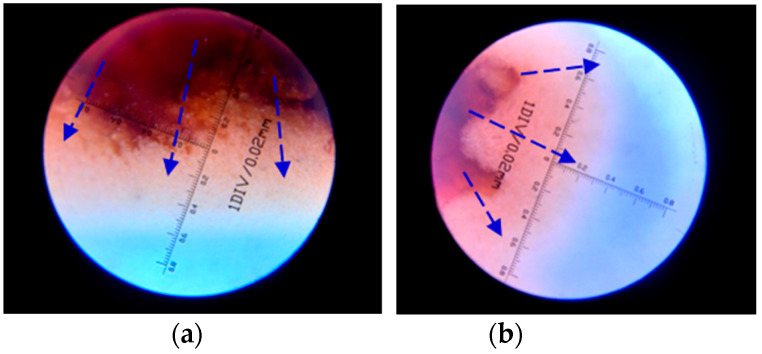
Water release trajectory of SAP in cement paste: (**a**) 3d; (**b**) 7d.

**Figure 13 polymers-15-03062-f013:**
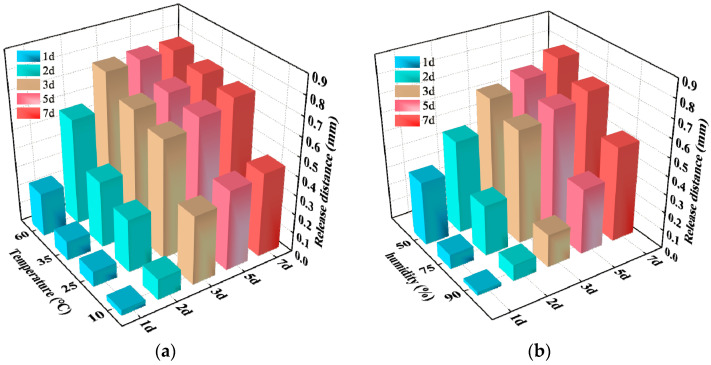
SAP water release distance test: (**a**) temperature; (**b**) relative humidity.

**Figure 14 polymers-15-03062-f014:**
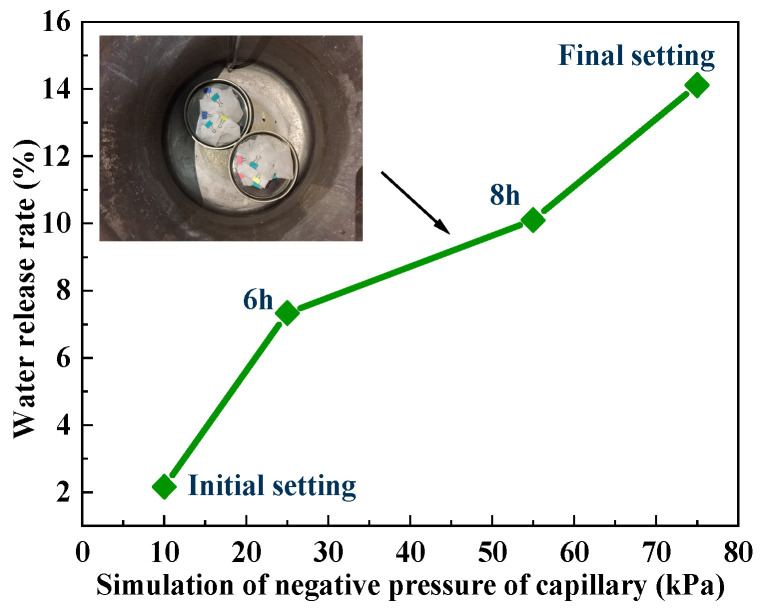
Water release rate under different simulated capillary negative pressure.

**Table 1 polymers-15-03062-t001:** The chemical composition of the cement (wt%).

Composition	SiO_2_	Al_2_O_3_	Fe_2_O_3_	CaO	MgO	SO_3_
Content	22.06	5.13	5.25	64.37	1.06	2.03

**Table 2 polymers-15-03062-t002:** Mix proportions of cement paste.

Strength Grade	*W/B*	Composition of Cement Slurry/(kg/m^3^)			
		Cement	Fly Ash	Water	Water Reducer
C30	0.37	370	65	160	2.61
C40	0.31	450	50	155	3.00

**Table 3 polymers-15-03062-t003:** Ion concentration determined in the cement paste (mol/dm^3^).

Solution Type	[K^+^]	[Ca^2+^]	[Na^+^]	[OH^−^]	[SO_4_^2−^]
0.37–10%	0.3507	0.0594	0.0392	0.1365	0.0910
0.37–15%	0.3147	0.0567	0.0383	0.0647	0.0431
0.37–20%	0.2756	0.0541	0.0357	0.0495	0.0330
0.37–25%	0.1915	0.0461	0.0326	0.0239	0.0159
0.31–10%	0.4185	0.0709	0.0468	0.2914	0.1943
0.31–15%	0.3756	0.0677	0.0458	0.1195	0.0797
0.31–20%	0.3290	0.0646	0.0427	0.0660	0.0440
0.31–25%	0.2286	0.0550	0.0389	0.0364	0.0243

**Table 4 polymers-15-03062-t004:** Mix proportion of simulated cement paste solution.

Solution Type	Chemical Reagent Ratio/(mmol/dm^3^)				
	CaSO_4_	K_2_SO_4_	Na_2_SO_4_	KOH	NaOH
0.37–10%	59	100	10	150	19
0.37–15%	57	90	10	135	18
0.37–20%	54	80	10	115	16
0.37–25%	46	60	10	70	13
0.31–10%	70	100	10	220	27
0.31–15%	68	90	10	195	26
0.31–20%	65	80	10	170	23
0.31–25%	55	60	10	110	19

**Table 5 polymers-15-03062-t005:** Absorption kinetic parameters of SAP at different temperatures and solutions.

Solution Type	Temperature	Parameter					R^2^
		*A*	*B*	*K_Q_*	*Q_t,eq_*	*Q_eq_*	
0.37–15%	10	0.042	0.032	25.245	31.056	30.594	0.998
	25	0.001	0.031	757.728	32.000	32.073	0.999
	35	0.011	0.029	74.559	34.014	33.657	0.999
	60	0.004	0.026	150.325	37.147	36.997	0.999
0.31–15%	10	0.075	0.031	14.695	31.706	30.122	0.998
	25	−0.011	0.034	99.732	28.670	29.262	0.999
	35	0.002	0.029	352.430	34.270	34.456	0.999
	60	0.013	0.027	62.633	36.153	34.805	0.997

**Table 6 polymers-15-03062-t006:** Swelling kinetic model of SAP in cement paste.

Solution Type	Sap Particle Size	Swelling Dynamic Rate Equation
0.37–15%	SAP-20	*Q*_t_= exp(1.805lnt−6.034)
	SAP-40	*Q*_t_= exp(0.965lnt−2.880)
	SAP-100	*Q*_t_= exp(1.157lnt−3.447)
0.31–15%	SAP-20	*Q*_t_= exp(1.004lnt−3.453)
	SAP-40	*Q*_t_= exp(0.397lnt−1.176)
	SAP-100	*Q*_t_= exp(0.262lnt−0.899)

## Data Availability

The data presented in this study are available on request from the corresponding author.

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
