# Peer review of "Research of Water Absorption and Release Mechanism of Superabsorbent Polymer in Cement Paste"

_polymers, 2023, doi:10.3390/polym15143062_

Round 1
Reviewer 1 Report
Dear authors,
In this manuscript,
Research of Water Absorption and Release Mechanism of 2 Super Absorbent Polymer in Cement Paste, polymers-2458776, the influence of environmental factors such as water-binder ratio, fly ash content, ionic valence, temperature and Super Absorbent Polymer (SAP) particle size on absorbance performance of SAP was studied. Based on the swelling kinetics equation and image analysis technology, the absorbance kinetics of SAP in cement paste was further studied. Moreover, the influence mechanism of temperature, humidity and capillary pressure on the water release behavior of SAP in cement paste was explored. Finally, the water absorbance-release mechanism of SAP in fresh cement paste was proposed.
Based on microscopic image technology, the dynamic swelling behavior of SAP particles was explored. The water release performance of SAP in cement paste was analyzed by both the tracer method and the negative pressure method (please, explain in!!!!!) . The results show that the water absorbance of SAP in cement paste varied from 27 to 33 times. The ionic valence had a significant effect on the water absorbance capacity of SAP, which could be embodied that the larger the ionic radius, the lower the absorbance of SAP. Based on the swelling kinetic equation of SAP and the time-dependent swelling morphology of SAP in cement paste, a swelling kinetic model was established. The water release performance of SAP was less affected by the capillary negative pressures, and it would not release the water prematurely during the plastic stage, which was conducive to the continuous internal curing process of hardened paste in the later age.
It is indicated that SAP can supplement the water of cement paste timely, increase the internal relative humidity and reduce the early microfracture caused by humidity shrinkage while the relative humidity is low. During the practical application of the project, the water release of SAP process is controlled to maintain a step-by-step to achieve continuous and stable internal curing.
Figs 1, 2, 5, 14-19, 21 are of poor quality. They need to be fixed.
My other comments are in the paper in pdf file,
Sincerely

Author Response
Point 1: tracer method and the negative pressure method (please, explain in!!!!!).
Response 1: Thank you for the reviewer’s comment and advice.
The tracer method refers to the use of red ink as a tracking marker to characterize the migration trajectory of internal curing water. The negative pressure method refers to the continuous test of the capillary pressure before the final setting of the cement paste by using the capillary pressure test device, and then based on the measured data, the vacuum equipment is used to apply pressure to the water release test of SAP, so as to evaluate the water release degree of SAP, the specific steps are shown in Lines 172-177.
Point 2: Mesh is not SI unit. It is need fix it in all of place where there is in the manuscript.
Response 2: Thank you for the reviewer’s comment and advice.
Following reviewer’s advice, annotations corresponding to international units of mesh are added in Lines 117-118.
Point 3: L116: This can be shown in a table. It's more transparent.
Response 3: Thank you for the reviewer’s comment and advice.
Following reviewer’s advice, show it as a table (Table 1) in Line 128.
Point 4: 4. L129: What is it PH? Did you mean pH?
Response 4: Thank you for the reviewer’s comment and advice.
It is a clerical error. PH in the document have been changed to pH in Line 134.
Point 5: Fig.3: The instruments on which you perform experiments do not need to be shown photographically. It is enough to specify the type of instrument, the manufacturer, the country where it was produced, as well as the basic conditions necessary for testing your samples.
Response 5: Thank you for the reviewer’s comment and advice.
Following reviewer’s advice, all the instrument pictures in the manuscript have been deleted.
Point 6: Table2: What is it NO.?
Response 6: Thank you for the reviewer’s comment and advice.
Following reviewer’s advice, NO. has been changed to Solution Type in Line 145.
Point 7: Table2: This is not the crystal clear what does mean?
Response 7: Thank you for the reviewer’s comment and advice.
Because the actual cement paste has a large consistency and is rich in a large number of particles, which affects the accuracy of the test, the chemical solutions are configured to simulate the actual cement paste.
Point 8: Table2: dm3!
Response 8: Thank you for the reviewer’s comment and advice.
Following reviewer’s advice, mmol/L has been changed to mmol/dm3 in Line 145.
Point 9: Fig.4: It is unnecessary to show pictures of the instruments. It has already been said how they should be presented.
Response 9: Thank you for the reviewer’s comment and advice.
Following reviewer’s advice, all the instrument pictures in the manuscript have been deleted.
Point 10: Fig.5: How were these images obtained?
Response 10: Thank you for the reviewer’s comment and advice.
These images were taken by camera when SAP absorbing red ink is put into white cement paste for water release test.
Point 11: 11. L169: Explain the method!
Response 11: Thank you for the reviewer’s comment and advice.
The negative pressure method refers to the continuous test of the capillary pressure before the final setting of the cement paste by using the capillary pressure test device, and then based on the measured data, the vacuum equipment is used to apply pressure to the water release test of SAP, so as to evaluate the water release degree of SAP. The specific steps are shown in Lines 172-177.
Point 12: Correct in all places where pH appears. Lines 244-271.
Response 12: Thank you for the reviewer’s comment and advice.
Following reviewer’s advice, all PH in the document have been changed to pH in Lines 247-275.
Point 13: Image of the instrument again. I have already said my opinion about the photos of the instruments. You have to agree with the Editor whether this is important data. As far as I'm concerned, is not.
Response 13: Thank you for the reviewer’s comment and advice.
Following reviewer’s advice, all the instrument pictures in the manuscript have been deleted.
Point 14: Table3: What is it? No?
Response 14: Thank you for the reviewer’s comment and advice.
Following reviewer’s advice, NO. has been changed to Solution Type in Line 349.
Point 15: Table3: Pay attention to significant digits. I think there are too many digits for these two parameters.
Response 15: Thank you for the reviewer’s comment and advice.
Following reviewer’s advice, the numbers of all parameters are retained at 3 digits after the decimal point in Line 349.
Point 16: Fig.13: Influence of SAP particle size, on what?
Response 16: Thank you for the reviewer’s comment and advice.
Following reviewer’s advice, replace this sentence with ‘The absorptance of SAP particle size in different solutions' in Line 371.
Point 17: The quality of the Figs 14-19 is very bad, How did you get them? The entire process is not easy to follow based on these Figs. How did you perform this experiment?
Response 17: Thank you for the reviewer’s comment and advice.
All these images are original images. And these images were captured dynamically during the experiment.
Point 18: Fig.17: Bad quality of Figs!
Response 18: Thank you for the reviewer’s comment and advice.
These images are original images.

Reviewer 2 Report
The paper investigates the water absorption process of Super Absorbent Polymer (SAP) in a simulated pore solution and the water release performance of SAP in cement paste. This study is both interesting and valuable; however, there are several aspects that require further clarification and revision.
(1) More detailed information regarding SAP should be provided, including its composition and synthesis methods.
(2) It would be beneficial to explain the rationale behind employing the teabag method for the water absorptance test of SAP.
(3) Regarding Fig. 9, it would be helpful to elucidate the reasons for selecting these specific pH values.
(4) It is essential to provide the mix proportion used in the water release distance tracer test.
(5) Could you please address whether the presence of red ink affects the release of water in SAP?
(6) In reference to L452, please provide an explanation as to why the distance over which SAP releases water increases with rising temperature.
(7) The conclusion should be refined and revised to highlight the innovative findings and significance of this research.

Author Response
Point 1: More detailed information regarding SAP should be provided, including its composition and synthesis methods.
Response 1: Thank you for the reviewer’s comment and advice.
Following reviewer’s advice, the composition and synthesis methods of SAP were increased in Lines 110-115: In the synthesis, deionized water was used as solvent, acrylic acid (C.P) was used as monomer, ammonium persulfate (A.R) was used as initiator, N, N-methylenebisacrylamide (C.P) was used as crosslinking agent, sodium hydroxide and sodium chloride (A.R) were used as neutralization agent, monomer concentration was 50%, neutralization degree was 70%. Acrylic acid was neutralized to sodium acrylate firstly, and then sodium acrylate was polymerized to sodium polyacrylate.
Point 2: It would be beneficial to explain the rationale behind employing the teabag method for the water absorptance test of SAP.
Response 2: Thank you for the reviewer’s comment and advice.
In the tea bag method, the tea bag mesh diameter we use can let cement paste enter and exit freely, and SAP will not come out, which can well test the water absorptance of SAP in cement paste.
Point 3: Regarding Fig. 9, it would be helpful to elucidate the reasons for selecting these specific pH values.
Response 3: Thank you for the reviewer’s comment and advice.
The reasons for selecting these specific pH values were the sentence of ’Previous experiments found that the pH of cement paste fluctuated between 12.30-12.55 before mixing to final gel, with an average of 12.43’ in Lines 248-249.
Point 4: It is essential to provide the mix proportion used in the water release distance tracer test.
Response 4: Thank you for the reviewer’s comment and advice.
Following reviewer’s advice, the sentences of ‘The W/B of white paste was 0.31 and the fly ash content was 15%’ was added in Lines 162-163.
Point 5: Could you please address whether the presence of red ink affects the release of water in SAP?
Response 5: Thank you for the reviewer’s comment and advice.
Red ink has little effect on the water release of SAP. The effect of red ink on the water release of SAP is minor. The preliminary test found that the water absorptance of using red ink is close to that of not using red ink. The use of red ink can help to more intuitively observe the migration trajectory of internal curing water.
Point 6: In reference to L452, please provide an explanation as to why the distance over which SAP releases water increases with rising temperature.
Response 6: Thank you for the reviewer’s comment and advice.
The reason that the distance over which SAP releases water increases with rising temperature is that curing temperature has a certain influence on the internal curing effect of SAP. The higher the temperature, the faster the water inside the cement paste evaporates, the greater the demand for water, and the more water release of SAP. The corresponding sentences were added in Lines 459-461.
Point 7: The conclusion should be refined and revised to highlight the innovative findings and significance of this research.
Response 7: Thank you for the reviewer’s comment and advice.
Following reviewer’s advice, some detailed information have been revised to refined the conclusion in Lines 508-522.

Reviewer 3 Report
The paper investigated the water absorption and release behavior of SAP in cement paste. The study is interesting and helpful for the application of SAP inside cement concrete. There are still some questions to be specified by the authors, which can help improving the quality of the paper before publication. This, a minor revision is needed.
1. The background of the research may need to be mentioned in Abstract.
2. L126: how to consider the difference between simulated solution and actual cement paste?
3. L159: explain why you chose white cement in SAP water release distance tracer test?
4. In Section 2.2, it is suggested to provide the results of ion concentrations in actual cement paste.
5. L244: please specify how you adjust the pH of the solution?
6. L370: how to determine the volume of SAP gel?
7. The first paragraph of introduction can be strengthened by some new and crack related studies, such as Comparison of fly ash, PVA fiber, MgO and shrinkage-reducing admixture on the frost resistance of face slab concrete via pore structural and fractal analysis; Influences of MgO and PVA fiber on the abrasion and cracking resistance, pore structure and fractal features of hydraulic concrete
7. L412: what is the basis of the description "The diffusion of solvents in SAP gels is mainly divided into three types"? Please elaborate.
8. The conclusion should be refined to highlight the innovation and the importance of this study.
fine
Author Response
Point 1: The background of the research may need to be mentioned in Abstract.
Response 1: Thank you for the reviewer’s comment and advice.
The background of the research has been mentioned in the abstract. The corresponding sentence is ‘The water absorption and release properties of Super Absorbent Polymer (SAP) internal curing agent are affected by many factors such as solution properties, surroundings temperature-humidity and the particle size of SAP, which determine the curing effect and the durability of cement concrete structures directly’ in Lines 12 to 15.
Point 2: L126: how to consider the difference between simulated solution and actual cement paste?
Response 2: Thank you for the reviewer’s comment and advice.
In the actual project, the cement paste will hydration in about 30 minutes, so the ion concentration of the cement paste at 30 minutes is tested several times, and then simulated.
Point 3: L159: explain why you chose white cement in SAP water release distance tracer test?
Response 3: Thank you for the reviewer’s comment and advice.
Because white cement can intuitively observe the water release path of SAP, black cement has poor visual effect, which will cause errors in the test.
Point 4: In Section 2.2, it is suggested to provide the results of ion concentrations in actual cement paste.
Response 4: Thank you for the reviewer’s comment and advice.
Following reviewer’s advice, the test results of ion concentration in actual cement paste were added in Line 144.
Point 5: L244: please specify how you adjust the pH of the solution?
Response 5: Thank you for the reviewer’s comment and advice.
Following reviewer’s advice, the sentences of ‘The pH was adjusted of the solution by adding deionized water and NaOH’ was added in Lines 250-251.
Point 6: 6. L370: how to determine the volume of SAP gel?
Response 6: Thank you for the reviewer’s comment and advice.
Following reviewer’s advice, the sentences of ‘The volume of SAP gel was determined by grid method, it was assumed that the SAP gel particles are spherical particles and the radius was the number of grids. The actual total length corresponding to the length direction of the image presented by the optical microscope was 1.0 mm, and the actual total width corresponding to the width direction was 0.73 mm’ was added in Lines 373-377.
Point 7: The first paragraph of introduction can be strengthened by some new and crack related studies, such as Comparison of fly ash, PVA fiber, MgO and shrinkage-reducing admixture on the frost resistance of face slab concrete via pore structural and fractal analysis; Influences of MgO and PVA fiber on the abrasion and cracking resistance, pore structure and fractal features of hydraulic concrete
Response 7: Thank you for the reviewer’s comment and advice.
I don’t think it has much to do with the content of the study.
Point 7: L412: what is the basis of the description "The diffusion of solvents in SAP gels is mainly divided into three types"? Please elaborate.
Response 7: Thank you for the reviewer’s comment and advice.
Following reviewer’s advice, a book ‘Super Absorbent Polymer’ is quoted as a reference in Line 418 and 589.
Point 8: The conclusion should be refined to highlight the innovation and the importance of this study.
Response 8: Thank you for the reviewer’s comment and advice.
Following reviewer’s advice, some detailed information have been revised to refined the conclusion in Lines 508-522.

Round 2
Reviewer 1 Report
Dear Editor,
Dear authors,
You answered most of my questions. I have a few more comments or questions.
1. How did you determine the influence of pH?
2. Your answer for figures 11-16 is that these are the original Figs. Nevertheless, I still think that such Figures (the quality at which they were obtained) should not be shown in a scientific journal.
3. The following was added in the conclusion: The water absorption of SAP in different simulated solutions was 27-33 times, it had a stable absorption performance. Explain!!!
4. The importance of this kind of research is not emphasized anywhere. Please explain that either in the introduction or at least in the conclusion.
Sincerely
Author Response
Point 1: How did you determine the influence of pH?
Response 1: Thank you for the reviewer’s comment and advice.
In this study, the pH of real cement paste was tested and determined by pH meter, the test results is shown in Table 1. It could be found from Table 1 that the pH of cement paste fluctuated in the range of 12.30-12.55 before final setting, with an average of 12.43. Based on these experimental datas, pH=12, pH=12.43 and pH=13 were chosen to explore the effect of pH on the water absorptance performance of SAP. During the water absorption test, it could be found that pH has obvious effect on the water absorptance and absorbed rate of SAP, which was detialed in Lines 256-275.
Table1 The pH of real cement paste
|
5min |
30min |
1h |
2h |
4h(initial setting) |
6h |
8h |
10h(final setting) |
|
12.30 |
12.37 |
12.33 |
12.43 |
12.48 |
12.55 |
12.52 |
12.48 |
Point 2: Your answer for figures 11-16 is that these are the original Figs. Nevertheless, I still think that such Figures (the quality at which they were obtained) should not be shown in a scientific journal.
Response 2: Thank you for the reviewer’s comment and advice.
Following reviewer’s advice, figure 11-16 were deleted and the corresponding part of the manuscript was modified in Lines 375-422.
Point 3: The following was added in the conclusion: The water absorption of SAP in different simulated solutions was 27-33 times, it had a stable absorption performance. Explain!!!
Response 3: Thank you for the reviewer’s comment and advice.
In fact, the water absorptance of SAP are sensitive to the property of solutions. The water absorptance of SAP in different solutions was fluctuated in the range of 20-500 times. For example, the water absorptance of SAP in deionized water is about 300-500 times, but it is only 30-50 times in NaCl solution, significant difference exits between the two solutions. However, the water absorptance of SAP in cement paste simulated solution was 27-33 times, so it was relatively stable. Therefore, it was considered that SAP had a stable absorption performance.
Point 4: The importance of this kind of research is not emphasized anywhere. Please explain that either in the introduction or at least in the conclusion.
Response 4: Thank you for the reviewer’s comment and advice.
Following reviewer’s advice, the sentences of ‘The calculation error of internal curing water was often caused by the inaccuracy of SAP water absorptance test in previous studies’ and ‘in order to control the water absorption and water release performance of SAP precisely’ which emphasizes the importance of this research was added in Lines 56-60.

Round 3
Reviewer 1 Report
Dear authors,
My comments are in pdf format of the paper. I believe that the manuscript can be published if you correct what I have suggested.
My recommendation to you is: take care of the clarity of what you wrote.
Sincerely

Author Response
Dear Reviewer
The modification again in the manuscript were in blue.
Point 1: Until the end of this sentence, replace the comma with ;
Response 1: Thank you for the reviewer’s comment and advice.
Following reviewer’s advice, replace the comma with ‘;’ in Lines 113-115.
Point 2: ‘A.R’ What is it?
Response 2: Thank you for the reviewer’s comment and advice.
'C.P' represents analytically pure, and 'A.R' represents chemically pure. Following reviewer’s advice, these have been deleted in Lines 113-116.
Point 3: Lines 118-119
Response 3: Thank you for the reviewer’s comment and advice.
Following reviewer’s advice, the format has been modified. in Lines 118-119.
Point 4: ‘u’ What is it?
Response 4: Thank you for the reviewer’s comment and advice.
It is a clerical error. ‘u’ has been changed to ‘μ’ in Lines 120-121.
Point 5: The name of the manufacturer should be provided.
Response 5: Thank you for the reviewer’s comment and advice.
Following reviewer’s advice, ‘ordinary Portland cement (PO.42.5) with density of 3.15 g/cm3 was used’ has been changed to ‘ordinary Portland cement (PO.42.5) produced by Guangdong Yingde Conch Cement Co., Ltd. was used with density of 3.15 g/cm3’ in Lines 124-125.
Point 6: In the table (which???) column with name Solution Type shows: the first number represent the water-binder ratio, while the second number represents percentage of fly ash...
Response 6: Thank you for the reviewer’s comment and advice.
Following reviewer’s advice, ‘In the table’ has been changed to ‘In the table 3 to 6’ in Line 145.
要点7:对于图 9 的标题来说,这还不够。
回应7: 感谢您的评论和建议。
根据审稿人的建议,“温度的影响:(a)~(b)0.37-15%;(c)~(d) 0.31-15%'改为“温度对SAP吸水能力的影响:(a)吸收率0.37-15%;(b) 吸收率为0.37-15%;(c) 吸收率为0.31-15%;(d) 0-31行的吸收率为15.330-331%。
